# Reversal of cancer gene expression correlates with drug efficacy and reveals therapeutic targets

Bin Chen[1,*], Li Ma[2,*], Hyojung Paik[1,3], Marina Sirota[1], Wei Wei[2], Mei-Sze Chua[2], Samuel So[2] & Atul J. Butte[1]

The decreasing cost of genomic technologies has enabled the molecular characterization of large-scale clinical disease samples and of molecular changes upon drug treatment in various disease models. Exploring methods to relate diseases to potentially efficacious drugs through various molecular features is critically important in the discovery of new therapeutics. Here we show that the potency of a drug to reverse cancer-associated gene expression changes positively correlates with that drug's efficacy in preclinical models of breast, liver and colon cancers. Using a systems-based approach, we predict four compounds showing high potency to reverse gene expression in liver cancer and validate that all four compounds are effective in five liver cancer cell lines. The *in vivo* efficacy of pyrvinium pamoate is further confirmed in a subcutaneous xenograft model. In conclusion, this systems-based approach may be complementary to the traditional target-based approach in connecting diseases to potentially efficacious drugs.

[1] Department of Pediatrics, Institute for Computational Health Sciences, University of California, San Francisco, 550 16th Street, San Francisco, California 94143, USA. [2] Department of Surgery, Asian Liver Center, School of Medicine, Stanford University, 1201 Welch Road, Stanford, California 94305, USA. [3] Biomedical HPC Technology Research Center, Korea Institute of Science and Technology Information, 245, Daehak-ro, Yuseong-gu, Daejeon 34141, South Korea. * These authors contributed equally to this work. Correspondence and requests for materials should be addressed to B.C. (email: bin.chen@ucsf.edu) or to M.S.C. (email: mchua@stanford.edu) or to A.J.B. (email: Atul.Butte@ucsf.edu).

Rapidly decreasing costs of molecular measurement technologies not only enable profiling of disease sample molecular features at different levels (for example, transcriptome, proteome and metabolome)[1–5], but also enable measuring of cellular signatures of individual drugs in clinically relevant models[6–9]. Exploring systematic approaches to find drugs for diseases through various molecular features is critically important in the discovery of new therapeutics. Among these molecular features, gene expression has been the most widely used[8]. The most commonly used approach starts with computing a disease gene expression signature—by comparing disease samples and control samples—followed by identifying drugs that have a reversal relationship with the disease signature. Although the majority of drug-induced gene expression experiments have been conducted in three cancer cell lines, this systems-based approach has led to the discovery of a number of drug candidates for various cancers (for example, small cell lung cancer[10], metastatic colorectal cancer[11], lung adenocarcinoma[12], Ewing's sarcoma[13] and renal cell cancer[14]), and remarkably even in non-cancer diseases (for example, inflammatory bowel disease[15] and osteoporosis[16]). A few computational analyses also demonstrated that this approach could recover a limited number of known drug indications[11,17,18]. However, each of the aforementioned studies evaluated this approach based on a very small set of tested drugs. None of the studies to date sought to explore the reversal relationship itself with drug efficacy systematically.

In this study, we analyse over 66,000 compound gene expression profiles from the Library of Integrated Network-based Cellular Signatures (LINCS) L1000 data set[9], more than 12 million compound activity measurements from ChEMBL[19], over 1,000 cancer cell line molecular profiles from the Cancer Cell Line Encyclopedia (CCLE)[20] and over 7,500 cancer patient samples from The Cancer Genome Atlas (TCGA)[21]. We quantify the reversal relationship between disease and drug gene expression signatures as the Reverse Gene Expression Score (RGES), a

measure of potency to reverse disease gene expression. We find that the RGES positively correlates with half-maximal inhibitory concentration ($IC_{50}$), a quantitative measure of drug efficacy often used to prioritize compounds in vitro. As a proof of principle, four compounds with significant RGES were newly identified as having efficacy against liver cancer, and each was successfully validated to exert antiproliferative effects against five liver cancer cell lines in vitro. Of these four compounds, pyrvinium pamoate, which had the lowest $IC_{50}$, was further validated to significantly reduce the growth of subcutaneous liver cancer cell xenografts in nude mice. This large-scale computational analysis demonstrates the feasibility and potential of investigating the potency to reverse disease gene expression as a tool for hypothesis generation in the drug discovery process.

## Results

**Disease gene expression signatures and RGES.** We created disease gene expression signatures from 7,514 samples across 14 cancers by comparing RNA-sequencing (RNA-Seq) gene expression from tumours and adjacent normal tissues, using data downloaded from TCGA. We then collected 66,612 compound gene expression profiles consisting of 12,442 distinct compounds profiled in 71 cell lines (with 83% of the measurements made primarily in 15 cell lines), using data downloaded from LINCS. Each profile involved the expression measurement of 978 genes, termed 'landmark genes'. The changes in the expression of these landmark genes were computed after compounds were tested in different concentrations (62% of the measurements were made in conditions under 10 μM) for 24 h (49%) or 6 h (51%; Supplementary Fig. 1). The computation of RGES was adapted from the previous Connectivity-Map (CMap) method[8] (Fig. 1a, see Methods). A lower negative RGES indicates higher likelihood to reverse disease gene expression and vice versa. We focused on three cancers, breast-invasive

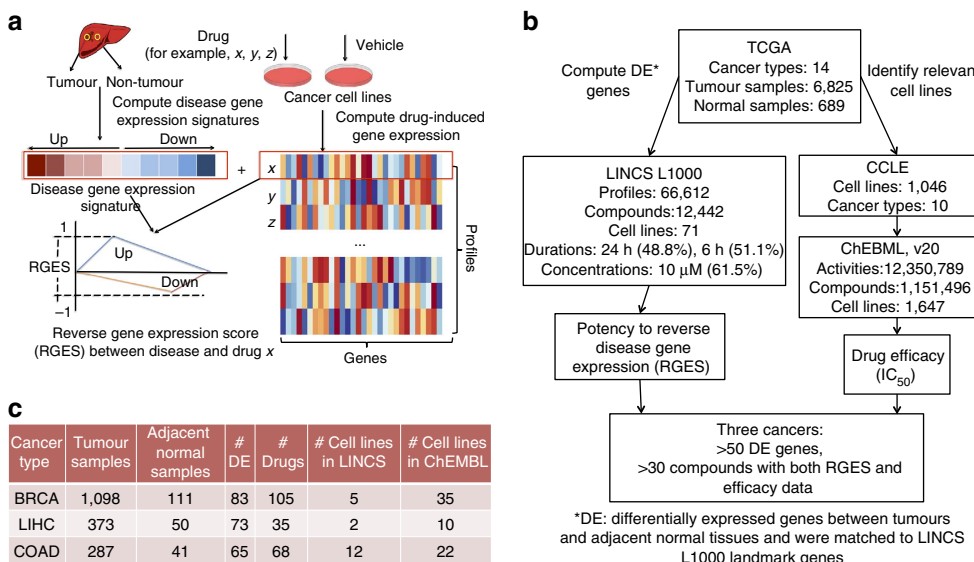

**Figure 1 | Retrieval of disease and drug gene expression profiles used for RGES computation.** (**a**) Schematic diagram of computing RGES based on the reversal relationship between disease and drug gene expression profiles. Lower RGES of a drug indicates higher potency to reverse disease gene expression. (**b**) Workflow of selecting appropriate cancer types to study. The public database TCGA was used to create cancer gene expression signatures; LINCS L1000 was used as the drug signature database; ChEMBL was used as the drug efficacy database; and CCLE was used to map cell lines among databases. Expression of the landmark genes was used by default in this study. The detailed method is described in Supplementary Methods. (**c**) Statistics of three selected cancers BRCA, LIHC and COAD. TCGA, The Cancer Genome Atlas; LINCS, Library of Integrated Network-based Cellular Signatures; CCLE, Cancer Cell Line Encyclopedia.

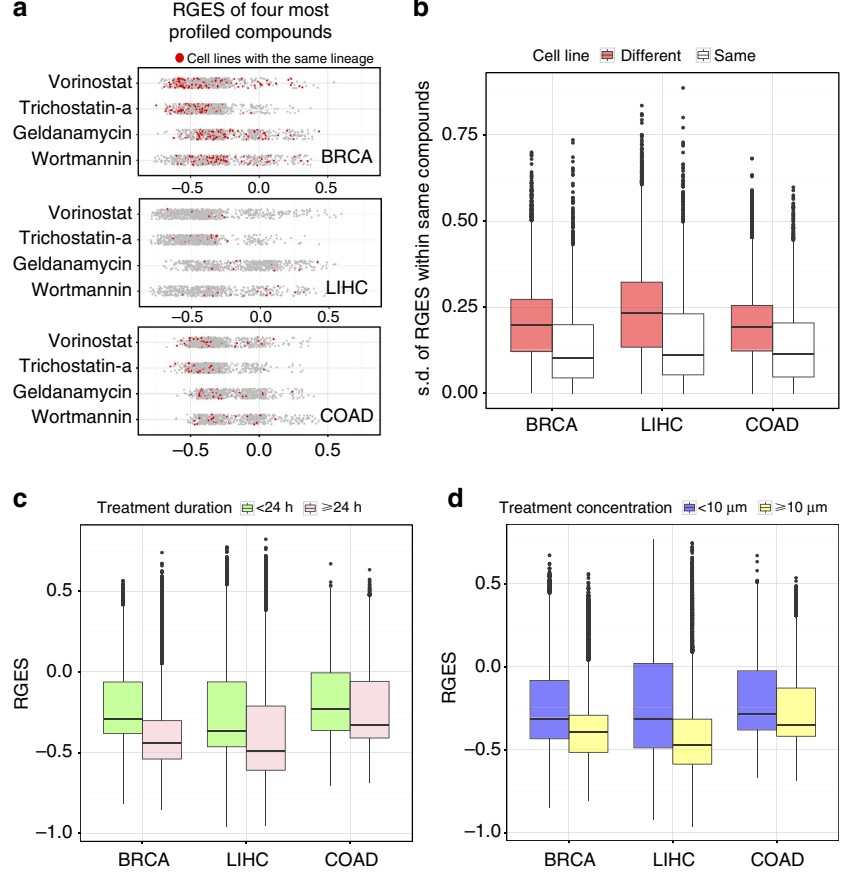

**Figure 2 | RGES is dependent on biological conditions.** (**a**) RGES of the top four most profiled compounds. Red dots represent drug signatures profiled in the cell lines that share the same lineage with the given cancer type. (**b**) s.d. of RGES of individual compounds across different cell lines versus across replicates within the same cell line. (**c**) RGES distribution between two treatment duration groups ($<24$ and $\geq 24$ h). (**d**) RGES distribution between two drug concentration groups ($<10$ and $\geq 10\,\mu$M). Treatment and concentration groups were categorized based on the distribution of drug profiles in LINCS.

carcinoma (abbreviated BRCA in TCGA), liver hepatocellular carcinoma (LIHC) and colon adenocarcinoma (COAD), based on the availability of relevant compound gene expression profiles on these cancers in LINCS and compound efficacy data on these cancers in ChEMBL[19] (Fig. 1b). These three cancer types have five (BRCA), two (LIHC) and 12 (COAD) cancer cell lines with the same cell lineage (Fig. 1c and Supplementary Data 1). After subsetting to the set of LINCS landmark genes, the signatures of BRCA, LIHC and COAD generated from the TCGA data included 83, 73 and 65 differentially expressed (DE) genes in tumours compared to the normal tissues of each, respectively (Fig. 1c and Supplementary Data 2). The signatures accurately classified tumours and adjacent normal tissues (Supplementary Fig. 2). These signatures were compared against all 66,612 compound gene expression profiles, resulting in one RGES for each compound profile for each cancer (scores range from $-1$ to 1, Supplementary Fig. 3a).

**RGES is dependent on biological conditions**. Each of the 12,442 compounds has an average of five measured gene expression profiles in LINCS. RGES of the same compound varies widely across different expression profiles regardless of cell line and lineage (Fig. 2a). For example, the most commonly profiled compound, vorinostat, has 860 measurements and scores between $-0.72$ and 0.54 with the median $-0.38$ across all five BRCA cell lines. We observed that the variations of RGES were greater

across different cell lines than within different replicates of the same cell line (when a compound was treated at the same concentration and duration; $P<2\times10^{-16}$, Fig. 2b). In addition, longer treatment durations ($\geq 24$ h) lead to lower RGES than shorter durations (when a compound was tested in the same cell line at the same concentration; $<24$ h; $P<2\times10^{-16}$, paired $t$-test, Fig. 2c). Likewise, higher drug concentrations ($\geq 10\,\mu$M) lead to lower RGES than lower concentrations ($<10\,\mu$M; when a compound was tested in the same cell line for the same duration; $P<1\times10^{-16}$, paired $t$-test, Fig. 2d). In addition, the variation of RGES of each compound may vary in different cancer types. For example, cytotoxic compounds ($n=9$) have less variation than targeted compounds ($n=69$) in BRCA ($P=0.003$), COAD ($P=0.004$), but not in LIHC ($P=0.09$). Collectively, RGES is dependent on cell line, drug concentration and treatment duration.

We examined three compounds (vorinostat, geldanamycin and gemcitabine) to further explore the effect of drug concentration on the RGES for predicting drugs in BRCA. We chose these compounds because they were tested under a sufficient number of distinct concentrations ($>15$) in the BRCA cell line MCF7. We computed their RGES against the BRCA signature generated from TCGA data, under different concentrations. For vorinostat, the correlation between RGES and drug concentration is $r=0.93$ ($P=1.1\times10^{-6}$) and $r=0.8$ ($P=4.5\times10^{-6}$) for 6 h treatment and 24 h treatment, respectively (Supplementary Fig. 4), suggesting that higher concentration of vorinostat

causes greater reversal of disease signature. For geldanamycin and gemcitabine, the correlation is significant ($P < 0.05$) when treated for 24 h, but not for 6 h (Supplementary Fig. 4). This indicates that some compounds may exhibit specific reversal potency only under certain concentrations and treatment durations.

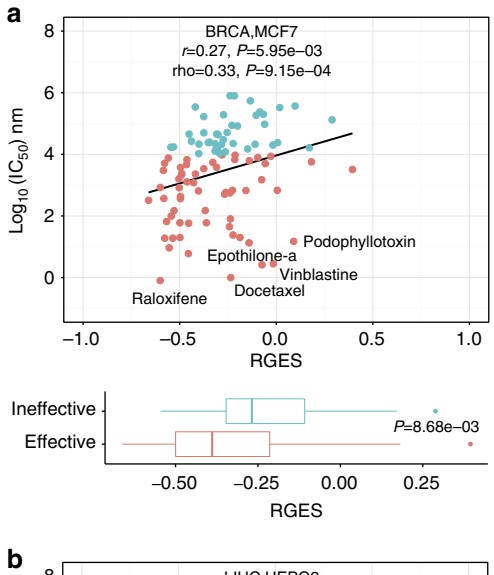

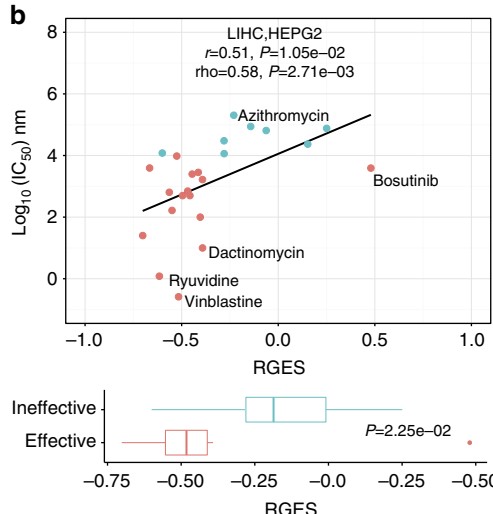

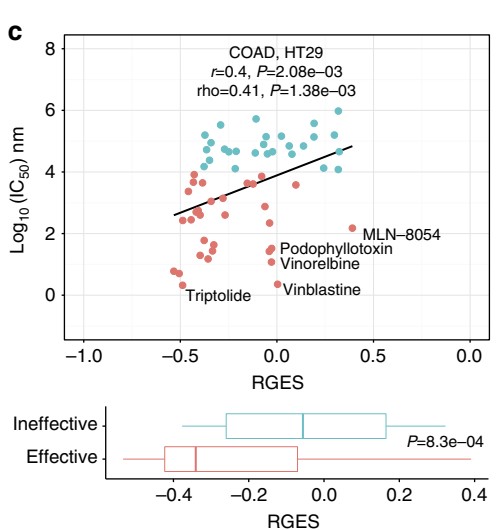

**RGES correlates with drug efficacy in cancer cell lines.** We next evaluated whether the RGES of a compound correlates with its efficacy in the same cell line. We chose cell lines that include the most number of compounds with both efficacy data in ChEMBL and gene expression profiles in LINCS for each cancer. As a result, we selected the cell lines MCF7 (100 compounds), HepG2 (24 compounds) and HT29 (58 compounds) for BRCA, LIHC and COAD, respectively. The median $IC_{50}$ was used when one compound was reported to have multiple $IC_{50}$ measurements in ChEMBL. Since RGES varies across drug concentrations and treatment durations, we set 10 μM and 24 h as a reference condition, and devised a method to normalize RGES from other conditions to this reference (see Methods). RGES and $IC_{50}$ have a strong correlation in all three cancer cell types after normalizing RGES (BRCA: Spearman correlation rho = 0.33, $P = 9.15 \times 10^{-4}$; LIHC: rho = 0.58, $P = 2.7 \times 10^{-3}$; COAD: rho = 0.41, $P = 1.38 \times 10^{-3}$; Fig. 3 and Supplementary Data 3). We further categorized compounds into a functionally effective group ($IC_{50} < 10$ μM) and a functionally ineffective group ($IC_{50} \geq 10$ μM). The functionally effective group of compounds presents with significantly lower RGES in three cancers (BRCA: $P = 8.68 \times 10^{-3}$; LIHC: $P = 2.25 \times 10^{-2}$; and COAD: $P = 8.3 \times 10^{-4}$, Student's $t$-test, Fig. 3). Among these cell lines, MCF7 is a model cell line for oestrogen receptor (ER)-positive BRCA. We thus created a signature for this specific subgroup from TCGA data, and computed RGES against this signature. We found that the correlation is retained in ER-positive subtype (rho = 0.37, $P = 5.63 \times 10^{-5}$, Supplementary Fig. 5a).

Our data suggest that reversal potency is correlated to drug efficacy in three cancer types, despite the large variations in RGES and $IC_{50}$. We also found that a number of effective compounds do not show potency to reverse disease gene expression (Fig. 3). For example, four microtubule inhibitors (docetaxel, vinblastine, podophyllotoxin and epothilone-α), and three microtubule inhibitors (vinblastine, podophyllotoxin and vinorelbine) do not show any potency to reverse gene expression in BRCA and COAD predictions, respectively. We further investigated the RGES of vinblastine, which has a whole-genome expression profile in MCF7 in an independent publicly available gene expression data set (GSE69845), and observed that vinblastine does not show any potency to reverse disease gene expression (RGES = 0.01, $P > 0.05$, Supplementary Methods) in that data set either.

**Summarized RGES for individual compounds across conditions.** One compound may have multiple available expression profiles due to its testing in various cell lines, drug concentrations, treatment durations or even different replicates, resulting in multiple RGES for one drug-disease prediction. Given these variations, we

**Figure 3 | Drug efficacy correlates with RGES in individual cancer cell lines.** (**a**) MCF7 for BRCA, (**b**) HepG2 for LIHC and (**c**) HT29 for COAD. In each cell line, only the compounds that have at least one expression profile and one $IC_{50}$ in that cell line were considered. Each dot represents one compound. Its RGES, for example, in plot A was computed by using BRCA gene expression and its expression profiled in MCF7 cells. RGES was normalized and merged when one compound has multiple expression profiles. Median $IC_{50}$ was used when one compound has multiple $IC_{50}$s from different studies. ANOVA and Spearman correlation were used to measure correlation between RGES and drug efficacy. The top five compounds with the most deviation from the linear line are highlighted. Boxplots illustrate RGES difference between effective compounds ($IC_{50} < 10$ μM) and ineffective compounds ($IC_{50} \geq 10$ μM).

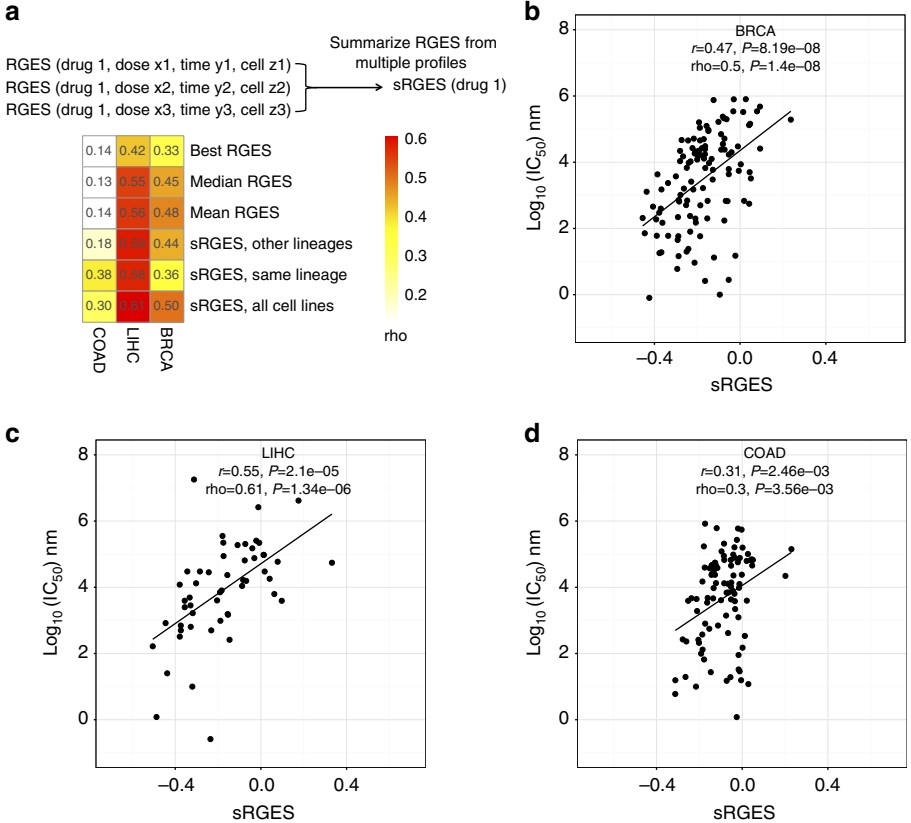

**Figure 4 | Drug efficacy correlates with sRGES from all cancer cell lines.** (**a**) Spearman correlations between sRGES and drug efficacy using different methods (best/median/mean of RGES, sRGES using cell lines from the same lineage, cell lines from different lineages or all cell lines). Correlations using other methods are listed in Supplementary Materials. Correlation between sRGES and drug efficacy in (**b**) BCRA, (**c**) LIHC and (**d**) COAD. ANOVA and Spearman correlation were used to measure correlation between sRGES and drug efficacy.

developed a summarization method to mitigate any bias and to compute a score that is representative of the overall reversal potency of a compound to a particular cancer, which we term summarized RGES (sRGES). Briefly, we set a reference condition (that is, 24 h and 10 μM) and estimated a new RGES if a RGES was not observed under the reference condition using a computational model. We also weighted the RGES by the degree of correlation between the gene expression profiles of the disease and the cell line in which the compound was tested. We tested a number of known methods to summarize scores and found that our method outperforms others (Methods, Supplementary Data 4). Of note, the approach that selects compounds based on the best RGES did not lead to a strong correlation (BRCA: rho = 0.33; LIHC: rho = 0.42; COAD: rho = 0.14, Fig. 4a), primarily owing to the large variation of RGES across different profiles. In comparison, our new summary score method led to the best correlation with drug efficacy (BRCA: rho = 0.50; LIHC: rho = 0.61; COAD: rho = 0.30; Fig. 4b–d and Supplementary Data 5). The correlation between sRGES and $IC_{50}$ remains significant in ER-positive BRCA (rho = 0.46, Supplementary Fig. 5b).

We also observed that the average correlation between sRGES and $IC_{50}$ decreases after excluding the profiles of the cell lines from the same lineage, but the correlation is still significant (BRCA: rho = 0.44; LIHC: rho = 0.59; COAD: rho = 0.18; Fig. 4a). This suggests that RGES that is solely based on gene expression profiles of the cancer cell lines from different lineages could still provide a reliable measure of drug reversal potency in these three cancers.

The efficacy of a drug may not be consistent across different studies[22] and can be measured by different matrices[23]; we thus sought to examine whether the correlation between reversal

potency and drug efficacy remains significant while using an external data set. We leveraged the recent large-scale pharmacogenomic database Cancer Therapeutic Response Portal (CTRP v2)[24], where responses of 860 cancer cell lines to treatment with each of the 481 compounds at various concentrations were quantified. Instead of $IC_{50}$, the area under concentration-response (AUC) was used to measure drug efficacy. In total, 192 compounds from LINCS were tested in 38 breast cancer cell lines, 22 liver cancer cell lines and 49 colon cancer cell lines. The median was used to summarize AUCs across multiple cell lines. The sRGES is still significantly correlated with the AUC in BRCA (rho = 0.47, $P = 6.9 \times 10^{-12}$), LIHC (rho = 0.43, $P = 7.1 \times 10^{-10}$) and COAD (rho = 0.36, $P = 3.67 \times 10^{-7}$; Supplementary Fig. 6). In addition, the growth rate (GR) inhibition metrics were recently proposed to be superior to conventional metrics ($IC_{50}$ and AUC) for assessing the effects of compounds in dividing cells[25]. We downloaded the GR data from the LINCS Pilot Phase Joint Project (http://www.grcalculator.org/), where the dose-dependent sensitivities of breast cancer cell lines were measured after the treatment of each of 107 compounds. Among them, 31 compounds have gene expression profiles in LINCS L1000. Although the number of compounds is small, we observed that the sRGES is still significantly correlated with the GR max in BRCA (rho = 0.38, $P = 0.03$; Supplementary Fig. 7).

**Experimental validation of drug hits for LIHC.** Since we observed that the sRGES significantly correlates with drug efficacy using existing public data, we next used this approach to identify novel compounds with high reversal potency for LIHC,

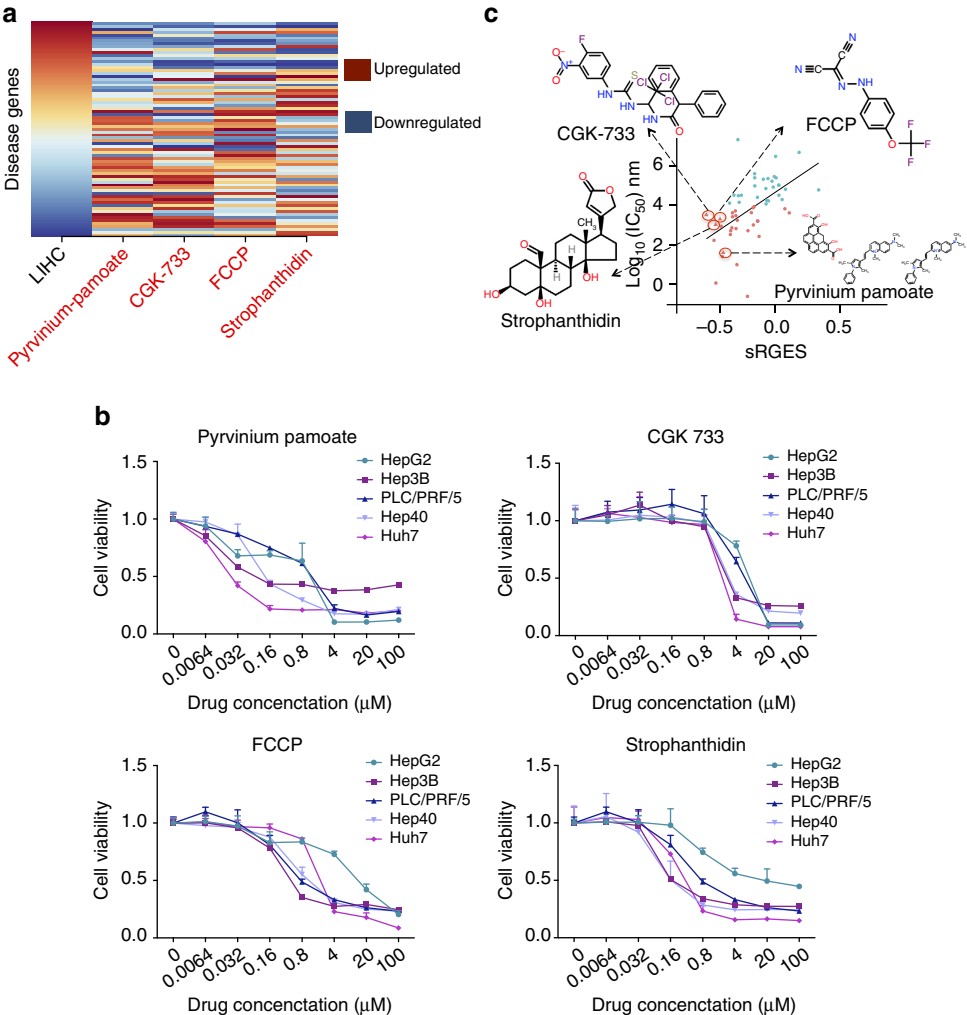

**Figure 5 | sRGES predicts drug efficacy in LIHC.** (**a**) Reversal relationship between LIHC gene expression and compound gene expression. The first column represents disease gene expression ranked by fold change. The remaining columns represent gene expression change after treatment with individual compounds. Compounds were selected based on their sRGES, and novelty in LIHC. When a compound has multiple profiles, the profile with the median RGES was selected for visualization in the heatmap. (**b**) Drug efficacy in five LIHC cell lines measured by cell proliferation assays after 72 h treatment. Data are shown as mean ± s.d. (**c**) Correlation between drug efficacy and sRGES after adding the four validated compounds. The median was used to merge drug efficacy across five cell lines.

a fatal malignancy with no effective treatment. Since only 1,845 compounds (15.8% of the tested chemical library in LINCS) were profiled in LIHC cell lines HepG2 and Huh7, the compound library would be limited if only these two cell lines were considered. We thus used sRGES to prioritize all compounds among the entire LINCS library consisting of over 66,000 compound profiles (Supplementary Data 6), and identified the top four compounds that have not been previously studied in LIHC: strophanthidin (sRGES: −0.49), carbonyl cyanide *p*-trifluoro-methoxyphenylhydrazone (FCCP, sRGES: −0.45), CGK-733 (sRGES: −0.56) and pyrvinium pamoate (sRGES: −0.42; Fig. 5a and Supplementary Data 6). Among these, CGK-733 was profiled in non-LIHC cell lines. Strophanthidin is a cardiac glycoside acting as an inhibitor of Na + /K +  ATPase[26]; FCCP is a mitochondrial uncoupler commonly used as a biological probe[27]; CGK-733 was originally defined as a selective inhibitor of the ataxia telangiectasia mutated (ATM) and the ATM- related (ATR) kinases[28]; and pyrvinium pamoate is an anthelmintic used to treat pinworm infection, and is also a Wnt signalling pathway inhibitor[29].

To test these four predicted candidate drug hits, we evaluated them for antiproliferative effects *in vitro* in a panel of five LIHC cell lines (HepG2, Huh7, Hep3B, PLC/PRF/5 and Hep40). The median $IC_{50}$s of strophanthidin, FCCP, CGK-733 and pyrvinium pamoate are 0.72, 1.78, 3.18 and 0.07 µM, respectively (Fig. 5b and Supplementary Table 1). When we annotated these four compounds on the plot using their median $IC_{50}$s in the five LIHC cell lines (Fig. 5b), we observed that they are close to our computed linear regression line (Fig. 5b). All four compounds have very different chemical structures (Fig. 5c) and primary mechanisms, yet they all show high likelihood to reverse LIHC gene expression and are effective in LIHC cell lines, indicating that sRGES can predict drug efficacy *in vitro*.

Further validation of pyrvinium pamoate, the compound with the lowest $IC_{50}$ value, showed that it significantly inhibited colony formation of HepG2 and Huh7 cells at 50 nM (close to its $IC_{50}$ value; Fig. 6a). Recent studies suggested that pyrvinium pamoate could inhibit the Wnt signalling pathway[29,30], which is frequently hyperactivated in LIHC[31]. Western blotting confirmed that Wnt pathway proteins (LRP6, Cyclin D1, Axin-1, Survivin) were inhibited in HepG2 and Huh7 cells (Fig. 6b). Using the TOPflash luciferase reporter assay, we demonstrated that pyrvinium pamoate indeed inhibited transcriptional activity

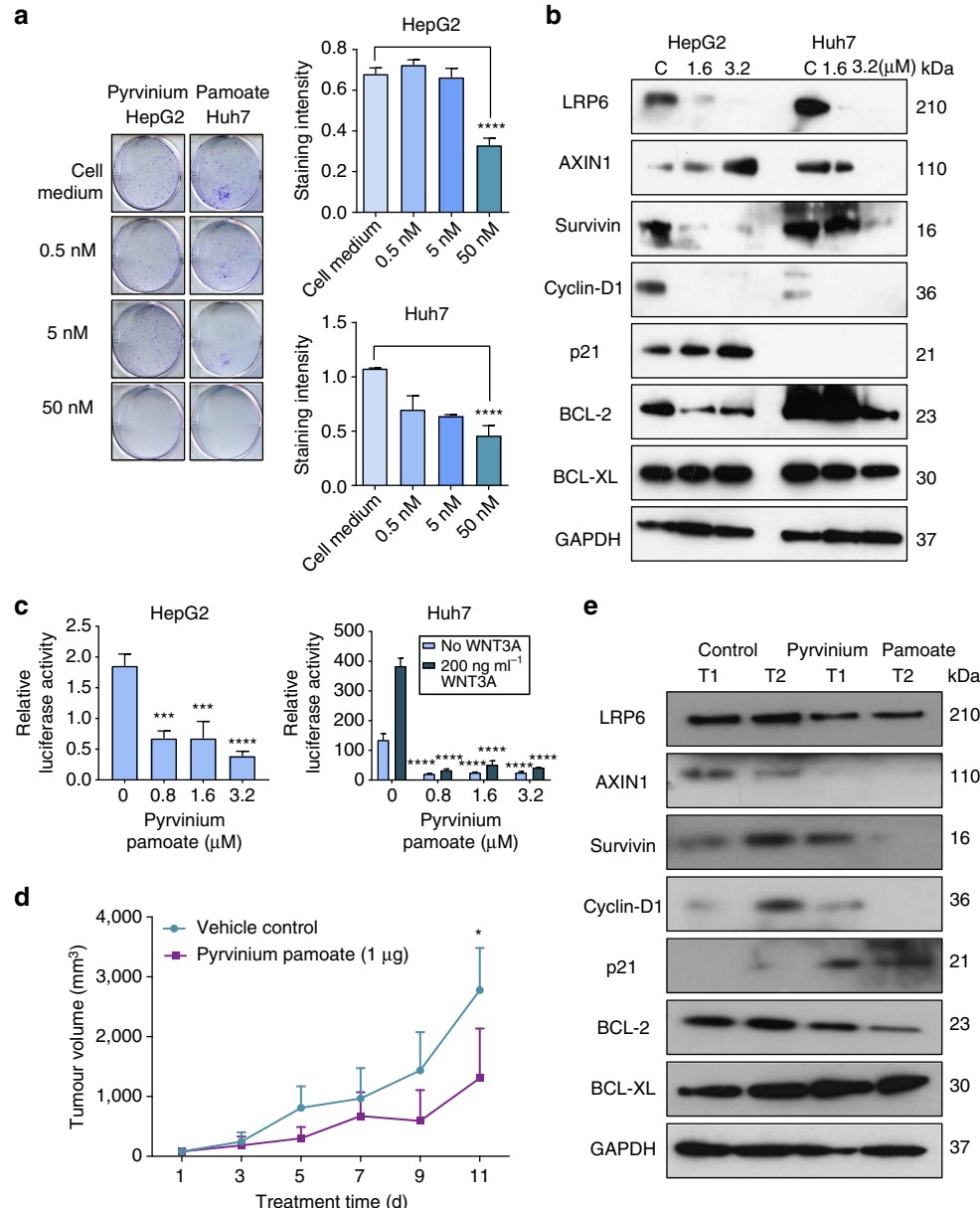

**Figure 6 | Pyrvinium pamoate inhibited growth of hepatocellular carcinoma *in vitro* and *in vivo* via targeting Wnt signaling. (a)** Pyrvinium pamoate inhibited colony formation. One-way ANOVA followed by Dunnett's multiple comparisons test (****$P = 0.0001$, 50 nM compared to cell medium). Data are shown as mean ± s.d. of three independent biological replicates. **(b)** Pyrvinium pamoate disrupted Wnt signalling proteins in HepG2 and Huh7 cells. Cells were treated for 24 h with DMSO (indicated as C), 1.6 μM or 3.2 μM of pyrvinium pamoate. **(c)** Pyrvinium pamoate attenuates β-catenin-mediated TOPflash activity in HepG2 and Huh7 cells. TOPflash and the Renilla luciferase reporter (control) were co-transfected into respective cell lines, and incubated with pyrvinium pamoate at the indicated concentrations, followed by 4 h incubation with or without rhWNT3A (200 ng ml$^{-1}$). Luciferase activity was normalized to Renilla concentrations in each sample and compared to vehicle control. For HepG2, one-way ANOVA followed by Dunnett's multiple comparisons test (***$P = 0.0001$, 0.8 μM compared to 0 as control; ****$P = 0.0001$, 1.6 μM, 3.2 μM compared to 0 as control). For Huh7, two-way ANOVA followed by Dunnett's multiple comparisons test (****$P = 0.0001$, 0.8, 1.6 and 3.2 μM compared to 0 as control). Data are shown as mean ± s.d. of three independent biological replicates. **(d)** Pyrvinium pamoate inhibited tumour growth in subcutaneous Huh7 xenografts *in vivo* ($n = 4$ per group). Data are shown as mean ± s.d. *$P = 0.035$ (Student's *t*-test). **(e)** Pyrvinium pamoate disrupted Wnt signalling proteins in subcutaneous Huh7 xenografts.

mediated through the Wnt/beta-catenin pathway (Fig. 6c). Intratumour administration of pyrvinium pamoate into subcutaneous Huh7 xenografts also significantly reduced the tumour volumes after 2 weeks of treatment, consistent with *in vitro* antitumour effects (Fig. 6d). Wnt pathway proteins were also inhibited in Huh7 xenografts (Fig. 6e), consistent with *in vitro* data. These data provided proof-of-concept that the sRGES can be used to accurately predict drug efficacy in *in vitro*

models of LIHC and that the drug hits will likely exhibit antitumour effects in *in vivo* models of LIHC as well.

**Reversal modules in each individual cancer.** Since sRGES evaluates compounds based on their overall reversal effect on a spectrum of cancer-associated genes, we next wanted to identify the genes that are specifically reversed by the effective

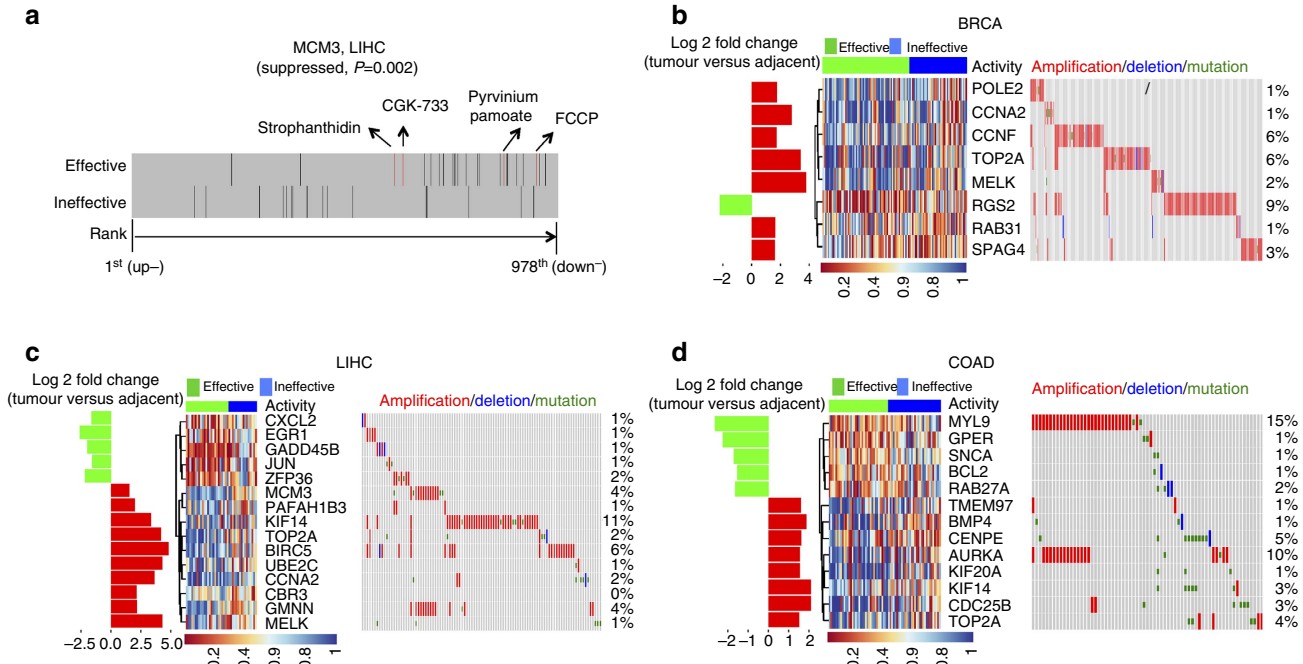

**Figure 7 | Genes reversed specifically by effective compounds. (a)** Example of *MCM3* suppressed by effective compounds in LIHC. Each line indicates the position of *MCM3* in a ranked drug expression profile. New effective compounds identified in this work are highlighted and coloured red. A lower rank suggests the gene is downregulated and a higher rank suggests the gene is upregulated by the corresponding compound. Reversal genes in **(b)** BRCA, **(c)** LIHC and **(d)** COAD. The heatmap in the middle indicates the relative position of a gene in a ranked drug expression profile. Positions are normalized and drug columns are ordered by its IC$_{50}$. Red shows the gene is upregulated and blue shows the gene is downregulated after treatment. The bar on the left shows their expression fold change between tumour samples and adjacent non-tumour samples, and the heatmap on the right shows their genetic alterations in TCGA patient samples obtained through cBioPortal[63].

compounds. We categorized compounds as functionally effective (IC$_{50}$ < 10 μM) or functionally ineffective (IC$_{50}$ ≥ 10 μM) for our three studied cancers, BRCA, LIHC and COAD; for LIHC, we also added the four compounds newly validated (described above) into the effective group. We then searched for post-treatment gene changes that would best discriminate the effective compounds from the ineffective compounds. As illustrated in Fig. 7a, expression of *MCM3*, which is upregulated in LIHC tumours compared to adjacent normal tissues, is suppressed specifically by effective compounds in LIHC. Using a leave-one-compound-out cross-validation approach to reduce over-fitting, we identified eight genes that were significantly reversed by the effective compounds in BRCA (Fig. 7b and Supplementary Data 7), including DNA Polymerase Epsilon Subunit B (*POLE2*) and Cyclin F (*CCNF*). Similarly, we identified 15 genes that were significantly reversed by effective compounds in LIHC (Fig. 7c and Supplementary Data 7), including Jun Proto-Oncogene (*JUN*) and Survivin (*BIRC5*); and 13 genes for COAD (Fig. 7d and Supplementary Data 7), including Aurora Kinase A (*AURKA*). These genes formed distinct reversal modules in three cancers, where genes were either suppressed or induced specifically by effective compounds.

In each of these three cancers, effective compounds exhibit the tendency to simultaneously suppress and induce multiple disease genes that are not frequently mutated but are related to disease progression and prognosis (Fig. 7b–d). For example, *BMP4* and *GPER* are suppressed and induced, respectively, by effective compounds in COAD. BMP signalling promotes the growth of primary human colon carcinomas *in vivo*[32] and activation of GPER exerts an inhibitory effect on colonic motility[33]. Overexpression of *RGS2*, a gene induced by effective compounds in BRCA, was reported to have an inhibitory effect

on BRCA cell growth[34]. *BIRC5*, the target of Wnt pathway, is also highly expressed in LIHC tumours and is suppressed by effective compounds in LIHC. Effective compounds including pyrvinium pamoate in LIHC induced the expression of *EGR1*. *EGR1* was shown to suppress cell growth and malignant phenotypes in LIHC[35] and was one of the immediate early genes repressed by Wnt signalling[36]. Together, this approach can potentially identify genes that may be associated with disease progression and that may be further investigated as therapeutic targets.

## Discussion
A primary goal of the precision medicine initiative is to identify new drugs for molecularly defined diseases[37]. The commonly used target-based drug discovery approach that focuses on interfering with individual targets is challenged by lack of drug efficacy, drug resistance and off-target effects[38–40]. The recent mixed results from the SHIVA trial, which selected therapies primarily based on actionable mutations, indicate that innovative approaches are going to be needed to increase the success of personalized medicine[41]. The approach of identifying drugs that reverse the molecular state of a disease may be a complementary method to the traditional target-based approach. Using gene expression as a representation of the molecular state, a number of studies have demonstrated its potential in drug discovery[12,42,43]; yet, there was no systematic way to correlate reversal potency and drug efficacy. Our study leveraged the emerging public cancer genomics and pharmacogenomics databases to address this challenge, and we successfully demonstrated that reversal potency correlates with drug efficacy and can be used to predict potential new drug candidates for several cancer types.

By integrating several large-scale public data sets, and using BRCA, LIHC and COAD as case studies, we demonstrated that drug reversal potency (as measured by its sRGES after drug treatment) is correlated to drug efficacy (measured by $IC_{50}$ after drug treatment), and yet the correlation is highly dependent on cell line, drug concentration and treatment duration. Taking these confounding factors into account using our summarization method could significantly improve the overall correlation between a drug's reversal potency and its efficacy. Even though cancer cell lines and patient tumour samples are different, a recent study demonstrated that cell lines faithfully recapitulate oncogenic alterations identified in tumours[44]. The positive correlation between the sRGES and $IC_{50}$ also indicates that combining disease gene expression derived from clinical samples and drug gene expression profiled *in vitro* could be a predictor of drug efficacy. Importantly, we showed that this correlation is retained even when the disease is not represented by cell lines of its own lineage in the drug expression databases. Since the 15 cancer cell lines primarily used in LINCS do not cover all cancer types, this finding suggests that our approach can be generalized to predict drugs for other cancers and cancer subtypes. As large volumes of drug gene expression profiles under different biological conditions can be produced owing to the rapidly decreasing costs of profiling technology, RGES can be used to screen compounds very efficiently and cost-effectively. More importantly, since RGES captures the molecular features of clinical samples, RGES is expected to be a clinically relevant predictor, compared to high-throughput screening technologies that measure drug activity in specific cell lines[20,24,45].

Our summarized RGES approach successfully predicted four novel compounds of distinct chemical structure and primary mechanisms, as being able to reverse LIHC gene expression. Independent validation of all four compounds using an *in vitro* antiproliferation assay in a panel of LIHC cell lines confirmed that all four compounds have potent antiproliferative effects in all five cell lines tested, providing proof-of-concept of our computational approach. Importantly, the most potent drug pyrvinium pamoate also significantly reduced the growth of subcutaneous xenografts of the LIHC cell line, Huh7, giving further confidence to our predictions. Pyrvinium pamoate, an FDA-approved drug for the treatment of pinworms, was reported to inhibit tumour growth via targeting Wnt signalling in breast cancer[30]. It was also reported to target the unfolded protein response[46], CD133 (ref. 47) and autophagy addiction[48]. The promising *in vitro* and *in vivo* antitumour effects of pyrvinium pamoate warrants a further investigation into its mechanisms and its potential use as a repurposed drug in liver cancer.

In addition to three primary cancers (BRCA, LIHC and COAD), we observed the significant correlation between reversal potency and drug efficacy in ER-positive BRCA, a subtype of breast cancer. We focused on this subtype, as it is the only subtype for which we could find sufficient activity and gene expression data for the computational analysis. It would be of great interest to explore this concept to identify drugs that reverse oncogenic pathway signatures, chemoresistance signatures or even individual patient signatures or to identify drug mimickers. Notably, it is likely that cytotoxic drugs or epigenetic inhibitors may present significant reversal potency to a variety of disease signatures; hence, additional work is needed to remove these nonspecific agents.

Besides its use to predict drug candidates, we demonstrated that the RGES could also be used to provide insights into the mechanisms of action of drug candidates. By studying the genes specifically reversed by effective drugs compared to ineffective drugs in each of the three cancer types studied, we found that each cancer has its own set of genes that were reversed (either induced or suppressed) by the effective drugs. These commonly reversed genes in each of these cancers may be common drug targets or downstream effector genes that mediate the antitumour effects of the effective drugs, and may be further investigated as potential therapeutic targets using traditional drug discovery approaches. Some of these reversed genes have also been reported to play functional roles in the progression of cancer, and our results suggest that interference with their expression (and therefore function) is associated with therapeutic outcome. Although the reversed genes are not frequently mutated in clinical tumour samples, it might be interesting to connect them to commonly observed mutations through biological pathways. Of note, this analysis was based on a limited number of landmark genes profiled in LINCS; it is possible that other relevant targets including those with high-frequent alterations could be identified with a greater coverage of transcripts in the drug expression profiles.

We also observed that the correlation of RGES to drug efficacy is not outstanding, and a number of compounds do not follow the trend. This may arise from several reasons. First, although $IC_{50}$ is one important measure of drug efficacy, it is known to vary across different studies. For example, in the COAD drug predictions, RGES of sorafenib reached as low as $-0.29$, but its $IC_{50}$ in HT29 cells is as high as 335 μM in ChEMBL, while its $IC_{50}$ in HT29 cells is 8 μM in another database[20]. It would be interesting to explore other matrices to measure drug efficacy[22,23]. Second, although the 978 landmark genes, which were primarily used to compute RGES, were carefully selected by the LINCS consortium, expression changes of these genes may not reflect the mechanisms of action of some compounds (for example, microtubule inhibitors). The performance could be improved if expression of the remaining genes could be inferred more accurately or other sequencing technologies such as RNA-Seq with more coverage of transcripts could be used. Third, since RGES depends highly on drug concentration and treatment duration, some compounds may need longer treatment durations or higher concentrations in order to exert downstream effects. For example, docetaxel presents different mechanisms under different doses[49]. We also note that RGES is derived from a profile after 6 or 24 h treatment, while $IC_{50}$ is derived from a response curve after 72 h or longer treatment. Lastly, it has been suggested that other types of data (for example, proteomics, metabolomics) should be integrated in computational drug discovery. For example, Niepel *et al.*[50] proposed that protein-signalling networks could be used to predict drug response in breast cancer cell lines, and Wei *et al.*[51] suggested that metabolomics could be used to predict response to neoadjuvant chemotherapy for breast cancer. Future improvements to the RGES method will aim to address some of these issues and to incorporate other types of omics data into its computation and subsequent drug prediction.

In summary, our computational approach provided a systematic way to connect disease gene expression with drug-induced expression profiles, and successfully demonstrated that drugs showing efficacy in cancer cells show enhanced potency to reverse disease gene expression compared to ineffective drugs. By analysing disease genes that are reversed by effective drugs, our approach can also provide possible insights into the disease development and drug mechanism. It also suggests the potential of using this computational method to assess the potency to reverse disease signatures composed by other molecular features (for example, protein) in addition to gene expression. Furthermore, since our method captures molecular features of patients, it is possible to develop signatures for individual patients, and use these to query and find more effective personalized treatments for individual patients. Lastly, since our summarized RGES approach provides one score for each drug based on its overall effect on a

given disease signature, further studies will investigate its predictive power in multiple models that reflect the heterogeneous subtypes of different cancers. Overall, our computational approach can be broadly applied to other cases where reliable gene expression data exist, and have the potential to speed up drug discovery in diseases with high unmet needs.

## Methods

**Data sets.** We collected RNA-Seq data of tumours and adjacent normal tissues from TCGA (http://firebrowse.org/)[21]. The transcripts, for which the number of tumour samples with raw count <1 is smaller than the number of normal samples, were removed. The tumour samples that are not correlated to the cancer cell lines from the same lineage in CCLE[20] were removed. The correlation between tumours and cell lines was computed by comparing their gene expression profiles[52,53]. Cell lines among LINCS, CCLE and ChEMBL were mapped using cell line name followed by manual inspection. Compounds between ChEMBL and LINCS were mapped using InChI keys. Compound $IC_{50}$s in cancer cell lines were retrieved from ChEMBL (version 20). Each $IC_{50}$ was manually inspected based on assay description. The assays where cell lines were manipulated, for example, to have acquired resistance to a drug, were ignored. As $IC_{50}$ of one compound may vary across different studies (Supplementary Fig. 8 and Supplementary Data 8) even in the same cell lines, we used the median to summarize the $IC_{50}$s. We also identified 9 cytotoxic compounds and 69 targeted compounds based on the previous pharmacogenomics study[44]. The details of data harmonization are provided in Supplementary Methods and in Supplementary Fig. 9.

**Disease/drug gene expression signatures.** RNA-Seq profiles were normalized and proceed using the R package DESeq V1 (ref. 54). Disease gene expression signatures were computed using the function nbinomTest (tumour samples versus unpaired adjacent tumour samples, log 2 fold change >1.5, adjusted $P<0.001$). Default parameters were used across all cancer types, unless specified. Drug gene expression profiles were downloaded from LINCS and processed as previously described[55]. Briefly, a full matrix composed by 476,251 signatures and 22,268 genes including 978 landmark genes was downloaded from the LINCS website (http://www.lincscloud.org/) as of September 2013. The meta-information of the signatures (for example, cell line, treatment duration, treatment concentration) was retrieved via the LINCS Application Program Interfaces (APIs; http://api.lincscloud.org/a2/). The signature, derived from the comparison of expressions between the samples treated with the perturbagen of interest and vehicle control, represents gene expression change upon treatment. The perturbagens can be small molecules, gene knock downs and gene overexpressions. Only small-molecule perturbagens with high-quality gene expression profiles (is_gold = 1, annotated in the meta-information) were further analysed.

**Disease selection.** In order to correlate RGES and $IC_{50}$, a sufficient number of DE genes (number of DE genes >50) and drug activities (number of drugs >30) are needed for each cancer type (Threshold selection was justified in Supplementary Methods and Supplementary Fig. 10). Among all the cancers available in TCGA, only BRCA, LIHC and COAD met the criteria (Supplementary Methods). In addition, ER-positive BRCA samples were identified from TCGA and their relevant cell lines were identified according to the literature[56,57].

**RGES computation.** The computation of RGES was modified from the connectivity score developed in previous studies[8,12]. Genes were first ranked by their expression values in each drug signature $i$. An enrichment score (es) of each set of up- and downregulated disease genes was computed based on their positions in the ranked list. Let $P$ be the total number of genes in the drug signature and let $m$ be the total number of up- or downregulated disease genes. First construct a vector $\mathbf{V}$ of the position $(1 \ldots n)$ for each of the genes in the disease signature on the basis of the values from the drug signature. Those were sorted in ascending order such that $V(j)$ is the position of disease gene $j$, where $j = 1,2,\ldots m$. Then, for each set of up- and downregulated disease genes, we computed $a_{up}$, $a_{down}$, $b_{up}$ and $b_{down}$ using the formulae provided in the Supplementary Material in Lamb et al.[8]. If $a_{up} > b_{up}$, we set enrichment score $es_{up} = a_{up}$, otherwise, $es_{up} = -b_{up}$. If $a_{down} > b_{down}$, we set enrichment score $es_{down} = a_{down}$, otherwise, $es_{down} = -b_{down}$. $es_{up}$ represents the absolute enrichment of an up gene list in a given profile and $es_{down}$ represents the absolute enrichment of a down gene list in a given profile.

In the previous studies, a connectivity score is set as 0 when both $es_{up}$ and $es_{down}$ are either positive or negative, leading to the enrichment of score 0 (Supplementary Fig. 3b). The highly enriched score 0 would bias the correlation analysis with drug efficacy (Supplementary Fig. 11). Therefore, we define $RGES = es_{up} - es_{down}$ regardless of the direction. Different from the connectivity score, RGES emphasizes the reversal correlation as it is aimed to capture the reversal relation between the disease and efficacious drugs. LINCS only profiled the expression of 1,000 landmark genes and imputed the expression of the rest of the genome using a computational model. We found that using landmark genes alone to compute RGES performs much better than including the imputed genes

(Supplementary Data 4). In order to exclude a possible artefact of limiting the disease signature genes to only the landmark genes, we performed a similar analysis using the Connectivity-Map data, where the whole-genome arrays in MCF7 were provided. We found that 31 compounds have gene expression profiles and drug efficacy in MCF7. The correlation between RGES and $IC_{50}$ is 0.52 ($P = 2.3 \times 10^{-3}$) while using the expression of the whole genome, and it decreased to 0.47 ($P = 6.7 \times 10^{-3}$) while using the expression of the landmark genes (Supplementary Fig. 12). Although new methods are being developed to impute gene expression[58], we primarily used the landmark genes in this study based on our analysis and previous studies[55,59].

In addition, we also computed Spearman, Pearson and Cosine similarity between disease and drug gene expression, which were suggested to be alternatives in computing the reversal relationship[60]. We performed similar analysis using these methods. We found that RGES led to the best correlation with drug efficacy (Supplementary Data 4). Therefore, in the following analysis, we decided to use RGES, computed based on the landmark genes.

**RGES summarization.** In previous studies using the CMap drug library, we and others chose the profile with the best likelihood to reverse disease gene expression[12,15]. Iorio et al.[43] developed a computational method to merge multiple gene expression profiles of one drug into a single profile and then compared each individual merged profile with the disease gene expression signature. These methods may not be directly applied to the new library LINCS L1000, where the drug gene expression profiles are much more diverse in terms of cell lines and treatment conditions than those in CMap. Existing tools such as L1000CDS[2] (ref. 61) being developed by the LINCS consortium currently only provide the scores of individual profiles for a given disease signature, without ranking the overall reversal potency of individual drugs. In the LINCS cloud (http://apps.lincscloud.org/), the mean connectivity score across multiple cell lines in which the perturbagen connected most strongly to the query was used to summarize connectivity scores. As drugs with a longer treatment and a higher concentration tend to present higher reversal potency (Fig. 2), it is not reasonable to compare the reversal potency of one drug under one treatment condition with another drug under a different treatment condition. Therefore, we developed a method to normalize RGES from other conditions to a reference condition, such that all normalized RGES could be compared under the same condition.

A common condition in LINCS is 10 μM concentration and 24 h treatment (accounting for 27% of all profiles)—we set this condition as a reference and any other conditions as a target condition. The majority of target conditions are with concentration <10 μM and treatment duration <24 h. Some drugs may have profiles under both conditions, while some may have profiles only under the target condition. We used these drugs, which were profiled in the same cell line with at least one target condition and at least one reference condition, to train a model. We assumed that the difference in RGES between a target condition and the reference condition is mainly dependent on its dose and time. We used $f(dose(i), time(i))$ to denote the difference, and the following formula to summarize RGES:

$$sRGES = \sum_{i}^{N}(RGES(i) + f(dose(i), \ time(i))) \times w(i)/N \qquad (1)$$

$$w(i) = cor(cell(i), tumours) \Big/ \max_{k} cor(cell(k), \ tumours), \qquad (2)$$

where $N$ is the number of drug profiles. $f(dose(i), time(i))$ was estimated by a computational model. Correlation between cell($i$) and tumour samples was estimated as the average of correlations between the cell line and individual tumours. The maximum correlation between cell lines and tumour samples was used to normalize correlation. The details of the model and its comparison with the methods being used in the LINCS cloud are described in the Supplementary Methods and Supplementary Fig. 13.

**Identification of reversed genes.** We first retrieved drug gene expression profiles and drug efficacy data ($IC_{50}$) from the cell lines that share the same lineage with a given cancer type. For those with multiple $IC_{50}$s, we chose their median $IC_{50}$. For those with multiple gene expression profiles, we chose the profile with the median RGES. As a result, each drug has only one gene expression profile and one $IC_{50}$. Each profile was sorted by its expression values: upregulated genes were ranked high (or on the top), and downregulated genes were ranked low (or on the bottom). Let $R(i, j)$ be the position of a up-/downregulated disease gene $i$ in a ranked profile $j$. Compounds were categorized into two groups: effective ($IC_{50} < 10$ μM) and ineffective group ($IC_{50} \geq 10$ μM) based on their activity in the cell lines. We chose 10 μM as the activity threshold because compounds with activity greater than 10 μM in primary screenings are often of little interest to continue[62]. For the upregulated genes, we defined the reversal genes as those that were ranked lower in the effective group than the ineffective group. For the downregulated genes, we defined the reversal genes as those that were ranked higher in the effective group than the ineffective group. One-sided Mann–Whitney–Wilcoxon test was used to assess the difference of the ranked between two groups. The gene with an adjusted $P$ value less than 0.25 was considered as a reversal gene.

To find a list of robust reversed genes, we used a leave-one-compound-out approach. For each trial, one compound was removed and reversed genes were then identified using the approach described above. Only the genes that were significantly reversed in all trials were kept.

**Chemicals.** FCCP and CGK-733 were purchased from Abcam (Cambridge, MA), and pyrvinium pamoate was purchased from USP (Rockville, MD). These three compounds were dissolved in dimethyl sulfoxide (DMSO) at stock solutions of 10 mM. Strophanthidin was purchased from Sigma-Aldrich (St Louis, MO) and was dissolved in 100% ethanol at a stock solution of 10 mM. All stock solutions were stored at $-20\,°C$ for subsequent use in cell-based experiments.

**Culture of LIHC cell lines.** The LIHC cell lines HepG2, PLC/PRF/5 and Hep3B were obtained from American Type Culture Collection (Manassas, VA), and the Huh7 cell line was a gift from Dr Mark Kay (Stanford University, CA). Hep40 cell line was a gift from Dr Xin Chen (University of California, San Francisco, CA). The human LIHC cell lines were maintained at $37\,°C$ in a humidified atmosphere (5% $CO_2$) in the following media types, supplemented with 10% fetal bovine serum, $100\,\mu g\,ml^{-1}$ penicillin and $100\,\mu g\,ml^{-1}$ streptomycin: Eagle's Minimum Essential Media (for HepG2, Hep3B and PLC/PRF/5) and DMEM (for Huh7 and Hep40). All media and supplements were obtained from Invitrogen (Carlsbad, CA). All cell lines were authenticated by short tandem repeat profiling (John's Hopkins University), and were mycoplasma-free.

**Cell proliferation assay.** Each of the five LIHC cell lines was seeded in 96-well, clear bottom plates (BD Biosciences, Franklin Lakes, NJ), with 5,000 cells in 200 µl of growth media per well. The cells were then treated with respective compounds in fivefold serial dilutions ranging from 100 to 0.0064 µM. After treatment period of 72 h, the media and compounds were removed and replaced with 100 µl of fresh growth media and 20 µl of CellTiter-96 AQueous One Solution Reagent (Promega, Madison, WI). After incubation for 2–4 h, absorbance was measured at 490 nm using the Powerwave XS microplate spectrophotometer (BioTek, Winooski, VT). $IC_{50}s$ were calculated as an estimate of each compound's efficacy. Three independent experiments were done, each in triplicates.

**Colony formation assay.** Each of the LIHC cell lines was seeded in six-well plates, with 5,000 cells in 2 ml of media per well. The cells were then incubated with pyrvinium pamoate at 0.5, 5 or 50 nM for 10 days, until colonies became sufficiently large to quantify. The media and compounds were replaced on days 3, 6 and 9. On day 10, the cells were washed once with $1\times$ PBS, fixed in ice-cold methanol for 10 min and stained with 0.5% crystal violet (in 25% methanol) for 10 min at room temperature. After rinsing with double-distilled water and drying at room temperature, images of the colonies were obtained using an Epson scanner. Each treatment was evaluated in triplicates, and representative images are shown. For relative quantification of colony formation, we determined the colony staining intensity by solubilizing the cell-associated dye in DMSO and measuring absorbance (OD580) of the dye-DMSO solution in a Powerwave XS microplate spectrophotometer (BioTek, Winooski, VT).

**Xenograft mouse model and drug treatment.** Animal work was approved by the Administrative Panel on Laboratory Animal Care at Stanford University. Animal studies were carried out in compliance with all federal and local institutional rules for the conduct of animal experiments. To generate subcutaneous xenografts, $2\times10^6$ Huh7 cells were suspended in 100 µl of Dulbecco's PBS (Invitrogen); after mixing with 100 µl of Matrigel (Corning, NY), the mixture was injected subcutaneously near the right forelimb of 6-week-old female NOD scid gamma (NSG) mice (The Jackson Laboratory, Bar Harbor, ME). Mice were then randomized into two groups ($n = 4$ each), and given intratumour injections of 50 µl DMSO (vehicle control), or 1 µg (dissolved in 50 µl DMSO) of pyrvinium pamoate every 3 days for 2 weeks. Tumour size was measured with a caliper before each treatment time point, and the tumour volumes were calculated using the formula: $V = (W^2 \times L)/2$. The full versions of western blots of the *in vitro* and *in vivo* experiments are available in Supplementary Fig. 14.

**Statistical analyses.** All cell-based experiments were independently repeated at least three times, and data from representative experiments are shown. All quantitative data are reported as means ± s.d. Unpaired *t*-test was used to calculate statistical differences between control vehicle and treatment group. Differences between two or more experimental groups were analysed by one-way analysis of variance (ANOVA) or two-way ANOVA. *P* values of $<0.05$ were considered significant. No statistical method was used to predetermine sample size for all experiments (*in vitro* and *in vivo*). The investigators were not blinded to allocation for the *in vivo* experiments. All statistical analyses of validation results were carried out using GraphPad Prism (GraphPad Software, San Diego, CA), and all other computational analyses were carried out in R version 3.2.4.

**Code availability.** The code is available at https://github.com/Bin-Chen-Lab/RGES.

**Data availability.** The data necessary for the analysis are available at Synapse (synapse.org; syn6182429). The rest of the data supporting the conclusions of this study are available from the corresponding author.

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

## Acknowledgements

We would like to thank B. Oskotsky for IT support. The work was partially supported by the National Institute of General Medical Sciences of the National Institutes of Health under award number R01GM079719. The content is solely the responsibility of the authors and does not necessarily represent the official views of the National Institutes of Health.

## Author contributions

B.C. conceived and designed the study, and performed the computational analysis. L.M. performed the majority of validation experiments. W.W. provided technical assistance with animal experiments and *in vitro* validation. B.C., L.M. and M.S.C. wrote the manuscript. M.S., H.P. and A.B. provided guidance on data analysis and manuscript preparation. M.S.C., A.B. and S.S. supervised the study.

## Additional information

**Competing interests:** A.B. is a founder and scientific advisor to NuMedii Inc. M.S. is an advisor to twoXAR. The remaining authors declare no competing financial interests.

