## [Peer review file · Nature Communications]

Editorial Note: This manuscript has been previously reviewed at another journal that is not operating a transparent peer review scheme. This document only contains reviewer comments and rebuttal letters for versions considered at Nature Communications. Mentions of prior referee reports have been redacted.

Reviewers' comments:

Reviewer #3 (Remarks to the Author):

In this revised manuscript, the authors addressed my concerns basically and improved the manuscript accordingly. In summary, the concept of this work may be of interest to others in the community and the wider field, while the further interpretation on computational model and experimental validation are needed. My novel comments for the revision are as follows:

1. The drug gene signatures were obtained based on cancer cell lines, where the cancer gene signatures were obtained relied on solid tumors, although the three cancer type tumor and cell lines belong to the same cell lineage, still need much more evidences to support they have similar molecular features. For example, comparing the significant somatic mutations among them.
2. In the validation part, both IC50 measurement and western blot experiments were implemented on cancer cells in vitro, how could the author claim their sRGES can be used to accurately predict drug efficacy in vitro and in vivo.
3. Besides WNT pathway, PI3K-Akt pathway is crucial in cancer differentiation and proliferation. The western blot experiments may be needed to validate the proteins in PI3K-Akt pathway.
4. The reviewer still thinks that it would be better to consider the somatic mutation profile of those so called "reversal" genes. It may help us to further understand the roles of those genes in cancer treatment if the author would like use them as the potential therapeutic targets.

Reviewer #5 (Remarks to the Author):

Chen and colleagues present a systematic analysis to explore the relationships between drug efficacy and drug-induced gene expression profiles in cancer cell lines. Their systems-based approach validated the original Connectivity Map concept using large-scale public gene expression and drug efficacy datasets. The authors also validated some of the compounds predicted to be potent in liver cancer experimentally using both in vitro and xenograft model. While the approach proposed in this manuscript is not excitingly novel, their analyses provide comprehensive evidence for the use of Connectivity Map concept in drug discovery. I think a major contribution to the field is the improved method (sRGES) for summarizing drug-induced gene expression signatures across cell lines and different experimental conditions. However, there are some details in the computational analysis need to be improved or justified to make the conclusions more solid. In my opinion, the manuscript could become more impactful if the authors expand the scope to non-cancer diseases and explore mimicking effects of compounds.

Below are my comments on the authors' revision in response to the original Reviewer #2's remarks:

1. The authors suggest that only three types of cancer (BRCA, LIHC, and COAD) have sufficient gene expression and drug efficacy data for their analysis based on the criteria of >50 differentially expressed landmark genes and >30 compounds. Could the authors

justify the use of 50 as the cutoff for number of differentially expressed landmark genes? That said, are drug efficacy more difficult to predict for disease signatures with less differentially expressed genes?

2. The authors use drug efficacy data from CTRP as an additional large-scale validation to address Reviewer #2's criticism. I have one additional comment on the drug efficacy metrics. As the authors also noted the inconsistency of drug efficacy metrics such as IC50 and AUC across different studies, a recently developed drug efficacy metric GR metric (Hafner et al, Nature Methods, 2016) has been proved to be a superior method and they also elucidated the flaws in IC50 and AUC. The LINCS consortium has generated drug efficacy datasets with the GR metric (<http://www.grcalculator.org/grbrowser/>). A validation on the correlation between sRGES and drug efficacy measured by the GR metric would make the conclusion more convincing.

3. Regarding to the comment from Reviewer #2 about the use of 978 landmark genes as opposed to both the landmark and inferred genes, I find it still not convincing that using the combination of measured and inferred genes performs worse than 978 measured genes alone in correlating with drug efficacy. Could this be an artifact of limiting the disease signature genes to only the landmark genes? This point would be more convincing if the authors repeat this analysis on the original Connectivity Map gene expression data, where all genes are directly measured. Such analysis can also conclude it is the inaccuracy in the imputation of the LINCS L1000 data that is weakening the correlation with drug efficacy.

I have the following additional comments on this manuscript:

1. The authors use a simple awarding function to account for the differences in RGES between the reference condition (10 μ M and 24h) and any other conditions when computing sRGES for any compounds. The constants in this function are estimated by averaging the differences in RGES across all compounds. Doing so would require the assumption that all compounds induce the same magnitude of RGES changes with regards to different dosages and time. Could the authors justify the use of averaging in estimating the constants? How much does this awarding function contribute to the correlation between drug efficacy and the sRGES?

2. I find the observation that the correlation between drug efficacy and sRGES solely based on gene expression profiles from cell lines of irrelevant lineages highly encouraging. Because this would imply that the drug-induced gene expression signatures generated from cancer cell lines could be used to find potent drugs for non-cancer diseases. Could the authors examine whether such correlation holds in the context of non-cancer diseases?

3. The gene expression values in Fig. S3 appear to be standardized across columns (samples), shouldn't it be standardized across rows (genes) to visualize the differential expression of genes across samples?

4. In the "RGES summarization" section, "existing tools being developed by the LINCS consortium" should refer to L1000CDS2 (Duan et al, npj Systems Biology and Applications, 2016) to be more specific.

5. The word "LINCS" is misspelled in supplementary material page 5.

6. A few references of supplementary figures are not updated, e.g., supplementary material page 5.

We have colored the changes as green in the main text. Please see our point-to-point responses below.

Reviewer #3 (Remarks to the Author):

In this revised manuscript, the authors addressed my concerns basically and improved the manuscript accordingly. In summary, the concept of this work may be of interest to others in the community and the wider field, while the further interpretation on computational model and experimental validation are needed. My novel comments for the revision are as follows:

1. The drug gene signatures were obtained based on cancer cell lines, where the cancer gene signatures were obtained relied on solid tumors, although the three cancer type tumor and cell lines belong to the same cell lineage, still need much more evidences to support they have similar molecular features. For example, comparing the significant somatic mutations among them.

--- We agree that more evidence is needed to support that cell lines and tumors have similar molecular features. In a recent study, Iorio *et al.* (2016, *Cell*) analyzed somatic mutations, copy number alterations, and hypermethylation across a total of 11,289 tumor samples and 1,001 cancer cell lines, and demonstrated that cell lines faithfully recapitulate oncogenic alterations identified in tumors. The samples in our study are a subset of their study. We have cited their work.

2. In the validation part, both IC50 measurement and western blot experiments were implemented on cancer cells in vitro, how could the author claim their sRGES can be used to accurately predict drug efficacy in vitro and in vivo.

--- We only validated the efficacy of one drug *in vivo*. In order to avoid any confusion, we have revised the text accordingly. We claimed that sRGES could be used to accurately predict drug efficacy in *in vitro* models of LIHC and the drug hits would likely exhibit anti-tumor effects in *in vivo* models of LIHC as well.

3. Besides WNT pathway, PI3K-Akt pathway is crucial in cancer differentiation and proliferation. The western blot experiments may be needed to validate the proteins in PI3K-Akt pathway.

---We agree that PI3K-Akt pathway is one of many pathways that are crucial in cancer differentiation and proliferation. In fact, Carrella *et al.* (*Oncotarget*, 2016) used a computational approach to identify pyrvinium pamoate as an inhibitor of PI3K-AKT pathway. In HCC, numerous other signaling modules are de-regulated, including some related to growth factor signaling (e.g., IGF, EGF, PDGF, FGF, HGF), cell differentiation (WNT, Hedgehog, Notch), and angiogenesis (VEGF). Intracellular mediators such as RAS and AKT/MTOR may also play a role in HCC development and progression (Moeini A, 2012, *Liver Cancer*). It is beyond the scope of this manuscript to validate the role of pyrvinium pamoate in all these pathways. We also recognize that pyrvinium pamoate may act through other targets and mechanisms (e.g., Venugopal, 2015, *Clinical Cancer Research*; Yu 2008, *PLOS ONE*; Deng, *Cell Death Dis.* 2013). It is encouraging that we identified a drug candidate with validated *in vitro* and *in vivo* anti-tumor effects, which warrants a further investigation into its mechanisms and its potential as a drug for treating liver cancer. We have discussed in page 15.

4. The reviewer still thinks that it would be better to consider the somatic mutation profile of those so called "reversal" genes. It may help us to further understand the roles of those genes in cancer treatment if the author would like use them as the potential therapeutic targets.

---We have added mutation data in Fig. 7. The reversal genes are not frequently mutated, which is not uncommon. For example, targets such as GPC3 and CDC37 are not known to be mutated in HCC and targets such as ER and HER2 are not known to be frequently mutated in breast cancer.

Reviewer #5 (Remarks to the Author):

Chen and colleagues present a systematic analysis to explore the relationships between drug efficacy and drug-induced gene expression profiles in cancer cell lines. Their systems-based approach validated the original Connectivity Map concept using large-scale public gene expression and drug efficacy datasets. The authors also validated some of the compounds predicted to be potent in liver cancer experimentally using both in vitro and xenograft model. While the approach proposed in this manuscript is not excitingly novel, their analyses provide comprehensive evidence for the use of Connectivity Map concept in drug discovery. I think a major contribution to the field is the improved method (sRGES) for summarizing drug-induced gene expression signatures across cell lines and different experimental conditions. However, there are some details in the computational analysis need to be improved or justified to make the conclusions more solid. In my opinion, the manuscript could become more impactful if the authors expand the scope to non-cancer diseases and explore mimicking effects of compounds.

Below are my comments on the authors' revision in response to the original Reviewer #2's remarks:

1. The authors suggest that only three types of cancer (BRCA, LIHC, and COAD) have sufficient gene expression and drug efficacy data for their analysis based on the criteria of >50 differentially expressed landmark genes and >30 compounds. Could the authors justify the use of 50 as the cutoff for number of differentially expressed landmark genes? That said, are drug efficacy more difficult to predict for disease signatures with less differentially expressed genes?

--- We learned the importance of the size of disease signatures from our previous studies, in which Connectivity Map was primarily used (Chen, *Gastroenterology*, 2017; Pessetto, *Ontotarget*, 2017). In the beginning of this work, we did try to evaluate this parameter. In our evaluation, we ranked disease genes based on fold change and selected a certain number of genes on each side (up/down) to build a disease signature. For each signature, we measured the correlation between RGES and IC₅₀ in individual cancer cell lines (BRCA: MCF7, COAD: HT29, LIHC: HepG2). We observed that as the size of gene set we chose was increasing, the correlation increased and then converged. In the three cancers, when the size of gene set of one side was about 25, the correlation did not increase (Supplementary Fig. 10). This indeed suggested that drug efficacy is more difficult to predict for disease signatures with less differentially expressed genes. Therefore, we only chose the diseases with > 50 differentially expressed genes. We note that this threshold is still arbitrary, as we did not take into account other factors (e.g., fold change, q-value). There may be an optimal combination of different parameters for each cancer. We did not intend to improve the correlation for each cancer. Other researchers may choose to improve the method further. We have justified the threshold selection in Supplementary Text.

2. The authors use drug efficacy data from CTRP as an additional large-scale validation to

address Reviewer #2's criticism. I have one additional comment on the drug efficacy metrics. As the authors also noted the inconsistency of drug efficacy metrics such as IC50 and AUC across different studies, a recently developed drug efficacy metric GR metric (Hafner et al, Nature Methods, 2016) has been proved to be a superior method and they also elucidated the flaws in IC50 and AUC. The LINCS consortium has generated drug efficacy datasets with the GR metric (<http://www.grcalculator.org/grbrowser/>). A validation on the correlation between sRGES and drug efficacy measured by the GR metric would make the conclusion more convincing.

---Thank you for introducing this method. We used the LINCS pilot set provided by this website. This dataset only includes breast cancer cell lines, so we used BRCA as an example. The authors suggested that GR50 and GR max are superior to conventional metrics for assessing the effects of small molecule drugs in dividing cells. In this pilot set, the dose-dependent sensitivities of breast cancer cell lines were measured after the treatment of each of 107 compounds. Among them, 31 compounds have gene expression profiles in LINCS L1000. However, some GR50 values were missing in the file; after communicating with the authors, we learned that GR50 is difficult to derive when the compounds are either too toxic or not effective. The authors suggested using GR max. We observed that the sRGES is still significantly correlated with the GR max in BRCA ($\rho = 0.38$, $p = 0.03$, Supplementary Fig. 7), yet not very remarkable. There are at least two reasons. First, the sample size is small (31 compounds from the GR website vs. 100 compounds from ChEMBL). Second, even though GR max is superior to conventional metrics for assessing the effects of small molecule drugs in dividing cells, it still does not address the inconsistency issue between different pharmacogenomics studies due to different standards in running assays. In this work, we used the drug efficacy data merged from multiple studies in ChEMBL. It might mitigate the inconsistency issue. We have discussed this on page 9 and 10.

3. Regarding to the comment from Reviewer #2 about the use of 978 landmark genes as opposed to both the landmark and inferred genes, I find it still not convincing that using the combination of measured and inferred genes performs worse than 978 measured genes alone in correlating with drug efficacy. Could this be an artifact of limiting the disease signature genes to only the landmark genes? This point would be make more convincing if the authors repeat this analysis on the original Connectivity Map gene expression data, where all genes are directly measured. Such analysis can also conclude it is the inaccuracy in the imputation of the LINCS L1000 data that is weakening the correlation with drug efficacy.

---Thank you for raising this concern. In order to exclude a possible artifact of limiting the disease signature genes to only the landmark genes, we performed a similar analysis using the Connectivity Map data, where the whole genome array in MCF7 cells is provided. We found that 31 compounds have gene expression profiles and drug efficacy data in MCF7 cells. The correlation between RGES and IC₅₀ decreased from 0.52 ($p = 2.3 \times 10^{-3}$) while using the original disease gene signature, to 0.47 ($p = 6.7 \times 10^{-3}$) while using a reduced gene signature (after mapping to 978 genes) (Supplementary Fig. 12). This suggested that the whole genome array might lead to a better correlation. In our previous evaluation of LINCS data (Chen, *CPT Pharmacometrics Syst Pharmacol.* 2015), we showed that the landmark genes led to a better result when correlating gene expression profiles to chemical structures. The recent CMap challenge also suggested that the current inference algorithm is effective but imperfect (<http://crowdsourcing.topcoder.com/cmap>). We have discussed this on page 21.

I have the following additional comments on this manuscript:

1. *The authors use a simple awarding function to account for the differences in RGEN between the reference condition (10 μ M and 24h) and any other conditions when computing sRGEN for any compounds. The constants in this function are estimated by averaging the differences in RGEN across all compounds. Doing so would require the assumption that all compounds induce the same magnitude of RGEN changes with regards to different dosages and time. Could the authors justify the use of averaging in estimating the constants? How much does this rewarding function contribute to the correlation between drug efficacy and the sRGEN?*

---This is a great point. In order to make compounds comparable, we need to compare their profiles under the same biological condition. But many compounds do not have profiles tested under the reference condition (10 μ M and 24 h); therefore, we need to use the data estimated from the population. Even if it is true that compounds may induce varying changes under different doses and durations, this simple awarding function estimated from a large number of compounds (>10,000) led to the dramatic increase in performance. For example, in COAD, the correlation increased from 0.14 (without the awarding function) to 0.30 (with the awarding function) (Fig. 4). In the current LINCS portal, different score metrics (e.g., mean_rankpt_2, rankpt_1) were developed to summarize connectivity scores. We also examined if their summarized scores are correlated to drug efficacy. In addition, we used the awarding function to summarize their connectivity scores, and compared our method with those in LINCS. When our method was used to summarize the scores, the correlation increased 37% (BRCA: 0.30 to 0.41), 6% (COAD: 0.35 to 0.37), and 12% (LIHC: 0.49 to 0.55) (Supplementary Fig.13). We have discussed this in Supplementary Materials page 6.

2. *I find the observation that the correlation between drug efficacy and sRGEN solely based on gene expression profiles from cell lines of irrelevant lineages highly encouraging. Because this would imply that the drug-induced gene expression signatures generated from cancer cell lines could be used to find potent drugs for non-cancer diseases. Could the authors examine whether such correlation holds in the context of non-cancer diseases?*

---It is a great idea to examine the correlation in non-cancer diseases, but it is beyond the scope of this manuscript. The drug efficacy (IC₅₀) data is derived from cell viability assays in cancer cells. Different metrics may be needed for other diseases. We hope to be able to study this in the future.

3. *The gene expression values in Fig. S3 appear to be standardized across columns (samples), shouldn't it be standardized across rows (genes) to visualize the differential expression of genes across samples?*

---Thanks for the suggestion. We have standardized the matrix across rows (genes).

4. *In the "RGEN summarization" section, "existing tools being developed by the LINCS consortium" should refer to L1000CDS2 (Duan et al, npj Systems Biology and Applications, 2016) to be more specific.*

---We have now cited this paper.

5. *The word "LINCS" is misspelled in supplementary material page 5.*

---We have corrected it.

6. *A few references of supplementary figures are not updated, e.g., supplementary material page 5.*

---We have updated the references.

REVIEWERS' COMMENTS:

Reviewer #5 (Remarks to the Author):

In the revised version of this manuscript, Chen and colleagues satisfactorily addressed my concerns. The reviewer appreciates the efforts made by the authors to perform analyses on additional datasets.

Reviewer #3 (Remarks to the Author):

The authors addressed our previous concerns basically.